# Hyperphosphatemia in Kidney Failure: Pathophysiology, Challenges, and Critical Role of Phosphorus Management

**DOI:** 10.3390/nu17091587

**Published:** 2025-05-05

**Authors:** Swetha Raju, Ramesh Saxena

**Affiliations:** Division of Nephrology, The University of Texas Southwestern Medical Center, Dallas, TX 75390, USA; swetharaju031@gmail.com

**Keywords:** hyperphosphatemia, phosphorus, kidney failure, chronic kidney disease, dietary management

## Abstract

Phosphorus is one of the most abundant minerals in the body and plays a critical role in numerous cellular and metabolic processes. Most of the phosphate is deposited in bones, 14% is present in soft tissues as various organic phosphates, and only 1% is found in extracellular space, mainly as inorganic phosphate. The plasma inorganic phosphate concentration is closely maintained between 2.5 and 4.5 mg/dL by intertwined interactions between fibroblast growth factor 23 (FGF-23), parathyroid hormone (PTH), and vitamin D, which tightly regulate the phosphate trafficking across the gastrointestinal tract, kidneys, and bones. Disruption of the strict hemostatic control of phosphate balance can lead to altered cellular and organ functions that are associated with high morbidity and mortality. In the past three decades, there has been a steady increase in the prevalence of kidney failure (KF) among populations. Individuals with KF have unacceptably high mortality, and well over half of deaths are related to cardiovascular disease. Abnormal phosphate metabolism is one of the major factors that is independently associated with vascular calcification and cardiovascular mortality in KF. In early stages of CKD, adaptive processes involving FGF-23, PTH, and vitamin D occur in response to dietary phosphate load to maintain plasma phosphate level in the normal range. However, as the CKD progresses, these adaptive events are unable to overcome phosphate retention from continued dietary phosphate intake and overt hyperphosphatemia ensues. As these hormonal imbalances and the associated adverse consequences are driven by the underlying hyperphosphatemic state in KF, it appears logical to strictly control serum phosphate. Conventional dialysis is inadequate in removing phosphate and most patients require dietary restrictions and pharmacologic interventions to manage hyperphosphatemia. However, diet control comes with many challenges with adherence and may place patients at risk for inadequate protein intake and malnutrition. Phosphate binders help to reduce phosphate levels but come with a sizable pill burden and high financial costs and are associated with poor adherence and psychosocial issues. Additionally, long-term use of binders may increase the risk of calcium, lanthanum, or iron overload or promote gastrointestinal side effects that exacerbate malnutrition and affect quality of life. Given the aforesaid challenges with phosphorus binders, novel therapies targeting small intestinal phosphate absorption pathways have been investigated. Recently, tenapanor, an agent that blocks paracellular absorption of phosphate via inhibition of enteric sodium–hydrogen exchanger-3 (NHE3) was approved for the treatment of hyperphosphatemia in KF. While various clinical tools are now available to manage hyperphosphatemia, there is a lack of convincing clinical data to demonstrate improvement in outcomes in KF with the lowering of phosphorus level. Conceivably, deleterious effects associated with hyperphosphatemia could be attributable to disruptions in phosphorus-sensing mechanisms and hormonal imbalance thereof. Further exploration of mechanisms that precisely control phosphorus sensing and regulation may facilitate development of strategies to diminish the deleterious effects of phosphorus load and improve overall outcomes in KF.

## 1. Introduction

Phosphorus is one of the most abundant minerals in the body and plays a pivotal role in numerous cellular processes, including maintenance of skeletal health, integrity of phospholipid bilayers, cell signaling, and synthesis of nucleic acids and adenosine triphosphate (ATP). The total amount of phosphorus in adults is approximately 700–800 g, comprising about 1% to 1.4% of fat-free mass [1]. About 85% of total phosphate is deposited in bone as hydroxyapatite crystals, 14% is present in soft tissues as various organic phosphates, and the remaining 1% is found in extracellular space as inorganic phosphate [2,3]. Of the phosphorus in the extracellular space, about 15–20% of the inorganic phosphorus is bound to plasma proteins, and the rest is complexed with sodium, magnesium, and calcium or circulates in blood as monohydrogen (HPO_4_^2−^) or dihydrogen forms (H_2_PO_4_^−^). The plasma inorganic phosphate concentration (commonly referred to plasma phosphorus concentration) is narrowly maintained between 2.5 and 4.5 mg/dL by intertwined interactions between fibroblast growth factor 23 (FGF-23), parathyroid hormone (PTH), and vitamin D that tightly regulate the precise balance in phosphate trafficking across the gastrointestinal tract, kidneys, and bones. Disruption of the tight hemostatic control of phosphorus balance can lead to altered cellular and organ functions that are associated with high morbidity and mortality [3].

In the past three decades, there has been a steady increase in the prevalent population of kidney failure (KF) needing kidney replacement therapies [4]. The individuals with KF have unacceptably high mortality (145.6 deaths per 1000 person–years, reported in 2022), and well over half (55.9%) of deaths are related to cardiovascular disease [4]. Abnormal phosphate metabolism is one of the major factors that is independently associated with vascular calcification and cardiovascular mortality in KF [3,5,6,7]. While overt hyperphosphatemia is observed late during the progression of chronic kidney disease (CKD), a series of adaptive processes involving FGF-23, PTH, and vitamin D occur in the early stages of CKD attempting to enhance phosphate excretion and maintain plasma phosphorus level in the normal range [8]. Recognizing the mechanism of phosphate homeostasis is paramount to comprehending the consequences of hyperphosphatemia and the rationale for its management in KF.

## 2. Phosphate Homeostasis

### 2.1. Phosphate Balance in Health

In the steady state, the amount of net intestinal uptake of dietary phosphorus equals the amount excreted by kidneys to maintain normal phosphorus balance [9,10,11] (Figure 1). The kidney and small intestine are the major organs involved in absorption, excretion, and reabsorption of phosphorus. Additionally, the balance between phosphorus inflow and outflow amid the extracellular fluid and bone and soft tissue is a contributory factor in the maintenance of normal serum phosphate concentration.

A typical Western diet contains about 1500 mg of phosphorus. Additionally, about 300 mg of phosphorus is exuded into the intestine via pancreatic and intestinal secretions. Approximately 1200 mg of ingested dietary phosphate is absorbed in the proximal intestine, giving a net daily phosphorus absorption of approximately 900 mg. Approximately 600 mg of phosphorus that is not absorbed in the intestine or released into the intestinal lumen eventually appears in the feces. Absorbed phosphorus enters the extracellular fluid pool, and about 300 mg of it moves in and out mainly of bone and, to a lesser extent, of soft tissues, maintaining a neutral balance. Approximately 900 mg of phosphorus, which is equivalent to the net amount absorbed from the intestine, is excreted in the urine (Figure 1) [12].

### 2.2. Intestinal Phosphorus Handling

Dietary phosphorus absorption occurs in the proximal small intestine, mainly jejunum, via two distinct pathways: the saturable transcellular pathway and the non-saturable paracellular pathway [11,13,14,15,16]. The transcellular pathway is the active intestinal phosphate transport involving sodium-dependent phosphate cotransporter IIb (NaPi-IIb, Npt2b), which is regulated by many factors, including dietary phosphorus intake and 1,25 dihydroxy vitamin D [17,18]. As a typical modern diet is abundant in phosphorus, the saturable transcellular pathway is quickly overwhelmed and does not remain the primary modality of intestinal phosphate absorption. Consequently, the paracellular pathway, which lacks the saturation limit, becomes the major route of phosphorus absorption in humans [19,20,21]. The paracellular phosphate absorption, driven by the phosphate concentration gradient, occurs passively through tight junction complexes claudins and occludins [15,21,22,23]. The permeability of phosphate through the paracellular pathway is influenced by the sodium–hydrogen exchanger 3 (NHE3), which is highly expressed in the small intestine [23,24]. Indeed, inhibition of enterocyte NHE3 by tenapanor has demonstrated reduced intestinal phosphorus absorption, resulting in a reduction in serum phosphorus level in patients with KF [23,25,26].

### 2.3. Renal Phosphate Handling

Kidneys are the primary organs responsible for phosphorus clearance and can excrete up to 4000 mg of phosphorus in a day. Following glomerular filtration, 80–85% of the filtered phosphate is reabsorbed in the proximal tubule (PT). Consequently, less than 15–20% of filtered phosphate is typically excreted in the urine in healthy individuals [27]. Renal phosphate absorption is facilitated by three sodium phosphorus cotransporters, namely sodium–phosphate co-transporter type IIa (NaPi-IIa, Npt2a), IIc (NaPi-IIc, Npt2c) and sodium–potassium co-transporter type III (Pit-2, Ram-1), all situated in the apical brush border membrane (BBM) of the PT [28]. These channels are regulated predominantly by FGF-23 and PTH.

FGF-23 is a 32 kDa glycoprotein, belonging to the endocrine FGF family, mainly produced by osteoblasts and osteocytes [29,30]. Circulating FGF-23 requires the co-receptor αKlotho to substantially enhance the binding affinity to FGF receptor 1c (FGFR1c). Binding of FGF-23 to FGFR1c leads to downstream signaling pathways, triggering internalization and degradation of NaPi-IIa and NaPi-IIc from the apical membrane, promoting increased urinary phosphate excretion. Furthermore, FGF-23 inhibits expression of 1α-hydroxylase (CYP27B1), the rate-limiting enzyme for vitamin D hormone (1,25(OH)2D3) synthesis. Both the phosphaturic and the 1,25(OH)2D3-lowering effect of FGF-23 protect against hyperphosphatemia by increasing urinary elimination of phosphate and reducing intestinal phosphate absorption, respectively [29,30] (Figure 2). In addition to FGF-23, PTH is a key phosphaturic hormone. PTH binds to PTHR1 receptor in PT, resulting in removal of NaPi-IIa and NaPi-IIc from the apical BBM via clathrin-coated pits [31,32]. However, unlike FGF-23, PTH increases transcriptional activation of 1α-hydroxylase (CYP27B1), thereby promoting 1α,25(OH)2D3) synthesis [30] (Figure 2).

### 2.4. Endocrine Regulation of Phosphate Homeostasis in Health and Kidney Disease

In normal individuals, a dietary phosphate load triggers the release of FGF-23 from bones. A recent study showed that PT glycolysis acts as a phosphate sensor in the kidney. In renal PT cells, phosphorus and not glucose is the rate-limiting step for glycolysis. Consequently, following dietary phosphorus load, renal phosphorus filtration and consequent absorption in PT promotes glycolysis, thereby increasing production of glycerol-3-phosphate (G-3-P). Subsequently, via a yet unknown mechanism, G-3-P is secreted into the circulation where it stimulates FGF-23 production in bone. FGF-23, in turn, downregulates renal tubule (RT) NaPi-IIa and NaPi-IIc, thereby reducing phosphate uptake, glycolytic flux, and, thus, further G-3-P production and so completes the negative feedback loop [9]. Additionally, by reducing 1,25(OH)2D levels, FGF-23 indirectly reduces intestinal phosphate absorption by downregulation of NaPi-IIb. Moreover, reduction of 1,25(OH)2D increases PTH release, which exerts its phosphaturic action by promoting recycling of NaPi-IIa and NaPi-IIc from apical BBM in PT [29,30,31].

In the early stages of CKD, a rising FGF-23 level following dietary phosphate load is still able to maintain a normal serum phosphorus level by enhancing fractional excretion of phosphorus (Figure 2). Moreover, FGF-23-mediated-reduction 1,25(OH)2D levels lead to secondary hyperparathyroidism, and the rise in PTH further augments renal phosphate excretion. However, as CKD advances, overt hyperphosphatemia ensues; as aforementioned, adaptive effects of FGF-23, PTH, and 1,25(OH)2D to enhance phosphate excretion are not able to overcome the phosphorus retention from continued dietary phosphorus intake and progressive reduction in glomerular filtration rate (GFR). In addition, other factors such as continued phosphate reabsorption by PT, decreased phosphate uptake into bone due to low 1,25(OH)2D levels, and ongoing phosphorus release from bone by elevated PTH further exacerbate hyperphosphatemia, thus creating a vicious cycle (Figure 2).

As discussed earlier, FGF-23 requires the co-receptor αKlotho to enhance the binding affinity to FGFR1c. αKlotho is a 130 kD transmembrane protein mainly present in the distal tubules of kidneys and choroid plexus of the brain. It comprises a very short intracellular domain, a single-pass transmembrane domain, and a large extracellular domain containing two internal repeats, KL1 and KL2 [33,34]. The extracellular domain of αKlotho protein is constitutively shed into circulation by the proteolytic cleavage as soluble αKlotho [33,34]. Additionally, a truncated secreted form of αKlotho, comprising only the KL1 domain, is generated by the alternative mRNA splicing of the αKlotho gene [33,34]. In the extracellular space, the soluble form is much more abundant, relative to the secreted form [33,34,35]. The membrane-bound αKlotho associates with FGFR1 to form a high affinity binding site for FGF-23 and plays a critical role in vitamin D and phosphate homeostasis [33,34,35]. The expression of αKlotho is significantly reduced in patients with CKD [30,36]. Furthermore, in progressive CKD, the phosphate load initiates compensatory hormonal mechanisms comprising increasing FGF-23 expression, low serum levels of active vitamin D, and an increase in PTH levels, to maintain phosphate balance [30]. Both high FGF-23 and low active vitamin D levels further suppress αKlotho expression, thereby inducing resistance to FGF-23 action by reducing its affinity to bind to FGFR1 in RT [30]. A reduced response to FGF-23 in the kidneys would blunt its phosphaturic effect and enhance intestinal phosphate absorption due to the inability of FGF-23 to inhibit vitamin D activation, thus perpetuating hyperphosphatemia and its adverse consequences [34]. Indeed, mutation of the Klotho gene results in upregulation of NaPi cotransporter expression, which increases phosphorus reabsorption, leading to a significant elevation of serum phosphate levels and acute heart failure in mice. Interestingly, normalization of serum phosphorus levels via a low-phosphate diet effectively prevented impairment in cardiac function and increased the survival rate and lifespan in Klotho mutant mice [37].

In contrast to the transmembrane isoform, the function and mechanism of action of soluble αKlotho are less clear. It is quite likely that soluble αKlotho acts as a hormone to regulate functions in tissues or cells that do not express FGFR [33,34,35]. For instance, soluble αKlotho can increase the abundance of both the transient receptor potential cation channel subfamily V member 5 (TRPV5) and renal outer medullary potassium channel (ROMK) 1, which are important calcium and potassium reabsorption channels in the kidney’s epithelial cell membrane [38,39,40].

## 3. Complications and Outcomes Associated with Hyperphosphatemia

Hyperphosphatemia is a common complication of advanced kidney disease [41]. While the phosphate level remains in the normal range in early stages of CKD, disturbances in phosphate metabolism occur early, triggering adaptive mechanisms to increase fractional excretion of urinary phosphate and reduce intestinal phosphate absorption. As the dietary intake overwhelms the excretory capacity of kidneys with progressive CKD, hyperphosphatemia ensues [42]. Hyperphosphatemia in CKD and KF has been associated with a high risk of cardiovascular disease (CVD), metabolic bone disease (MBD), and high cardiovascular (CV) and overall mortality [3,5,6,7,43,44,45]. Mortality from CVD in KF patients is about 20 times higher than that in the general population, and CVD accounts for more than half of the deaths in KF [4,46,47]. Furthermore, hyperphosphatemia is associated with a faster progression of CKD in non-dialysis CKD patients [48].

### 3.1. Hyperphosphatemia: Cardiovascular Risks and Mortality

Numerous studies have shown a strong association between hyperphosphatemia and CVD in CKD. Mechanistically, hyperphosphatemia is linked to an increased risk for CVD through multiple physiologic mechanisms. First, high phosphate concentrations and calcium phosphate products may increase vascular and soft tissue calcification [49,50]. While, to some extent, there is passive precipitation of calcium–phosphate in soft tissues, high extracellular phosphate induces the expression of osteoblastic genes in vascular smooth muscle cells (VSMCs), causing a transformation of VSMCs into osteoblast-like cells and fostering vascular calcification [51]. Moreover, hyperphosphatemia triggers remodeling of the extracellular matrix around VSMCs, promoting calcification in the medial layer of the vasculature [52,53,54]. In addition, phosphate exposure activates pro-inflammatory cellular signaling in VSMCs, thereby initiating oxidative stress and causing DNA damage that can lead to more inflammation and vascular calcification [55,56].

Secondly, phosphate retention raises FGF-23 and PTH concentrations. Both FGF-23 and PTH have been independently associated with direct pathogenic CV effects [57,58]. Increased FGF-23 levels have been demonstrated to enhance pro-inflammatory cytokines, which further worsens vascular calcification. Clinical and experimental studies have observed FGF-23 to be an independent risk factor for hypertension, left ventricular hypertrophy, congestive heart failure, and CV mortality [57,59,60,61,62]. Likewise, excess PTH is associated with pro-inflammatory effects, hypertension, impaired myocardial energy production, cardiac fibrosis, left ventricular hypertrophy, and heart failure [59,60,61,62,63,64,65,66,67,68].

Recent studies have shown that soluble αKlotho possess renoprotective and vasoprotective effects [36]. It is plausible that reduction in αKlotho expression in CKD can be associated with adverse consequences. Indeed, experimental data demonstrate that αKlotho deficiency accentuates vascular oxidative stress that is ameliorated by in vivo klotho gene delivery [69,70]. Moreover, transgenic mice that overexpress klotho exhibit lower oxidative stress [71]. Furthermore, the mutation of the Klotho gene results in upregulation of NaPi cotransporter expression, which increases phosphorus reabsorption, leading to significant elevation of serum phosphate levels and acute heart failure in mice. Interestingly, normalization of serum phosphorus levels by low phosphate diet effectively prevented impairment in cardiac function and increased the survival rate and lifespan in Klotho mutant mice [37]. Data from human studies, on the other hand, reveal conflicting results. For instance, in a recent systemic review and meta-analysis, CKD patients with low klotho levels were associated with significantly high all-cause and CV mortality compared to those with high klotho levels. Furthermore, the risk of CKD progression and KF with need for KRT was significantly higher in patients with lower klotho levels [72]. On the contrary, in a large, diverse, prospective chronic renal insufficiency cohort (CRIC) study, 5-year survival, heart failure, hospitalization, atherosclerotic CV events, and CKD progression did not differ between high- and low-Klotho groups. In contrast, FGF-23 was significantly associated with mortality and heart failure hospitalization independent of Klotho levels [73]. In summary, most of the studies on Klotho have been performed in animal models and have advanced the knowledge in understanding the role of αKlotho in regulating key physiological processes, including aging, phosphate metabolism, and vascular health. However, many aspects of the mechanisms by which αKlotho regulates various biological activities and their clinical significance remain to be elucidated.

### 3.2. Hyperphosphatemia and Risk of Mortality and Progression of Renal Disease

Various epidemiological studies have observed strong associations between high phosphorus levels and morbidity and mortality in CKD and KF patients [43]. A 2011 meta-analysis reported an 18% increased risk of death for every increase of 1 mg/dL in serum phosphate in CKD patients [45]. In another prospective cohort study, a serum phosphate level ≥3.5 mg/dL was associated with significantly increased mortality risk among CKD patients [74]. Furthermore, compared to CKD patients with normal phosphorus levels, the risk of death increased linearly with each subsequent 0.5 mg/dL increase in serum phosphate level and nearly doubled in patients with a serum phosphate level ≥4.5 mg/dL [74]. Yet, another study observed a 1.62-fold increase in mortality risk in CKD patients for every 1 mg/dL increase in serum phosphate [75]. Even in individuals with preserved kidney function, higher serum phosphate is associated with high mortality. In a post hoc analysis of the data from the Cholesterol And Recurrent Events (CARE) study, a graded independent relation between higher serum phosphate and the risk of death and cardiovascular events was observed [76]. Similarly, in a meta-analysis of 24 clinical trials in patients without CKD, serum phosphate was associated with higher mortality [77].

In addition to risk of mortality, higher serum phosphate concentrations are associated with a faster progression of CKD [48]. In experimental studies, phosphorus loading leads to kidney fibrosis in normal animals and promotes progression of kidney disease in CKD animals [78,79]. Likewise, in individuals without kidney disease, the risk of KF was significantly higher in patients with the highest-quartile phosphate compared to those with the lowest quartile, suggesting that relatively high phosphate levels within the normal range can be a risk factor for the development of CKD [80]. Among patients with CKD, those with higher serum phosphorus levels had a faster progression to KF than those with lower phosphate levels [81]. Furthermore, a meta-analysis of 12 cohort studies observed a 1.36 risk of KF and a 1.2-fold increase in mortality for every 1 mg/dL increase in serum phosphate levels in CKD patients [82].

### 3.3. Hyperphosphatemia and Metabolic Bone Disease

High phosphate is closely associated with metabolic bone disease (MBD), a widespread complication of CKD and KF. Even before the development of overt hyperphosphatemia, phosphorus dietary phosphorus load in CKD leads to an increase in FGF-23 and PTH, key factors driving the development of CKD-MBD [8,83]. High PTH causes further phosphate release from bone, worsening overall phosphorus balance and stimulating additional FGF23 and PTH production, leading to the development of high turnover bone disease, with a high risk of musculoskeletal pain and bone fractures [8,83]. Indeed, observational data noted a significant increase in bone fracture risk in hemodialysis patients with a baseline serum phosphate greater than 6.1 mg/dL [83]. Likewise, data from the Dialysis Outcomes and Practice Patterns Study (DOPPS) showed higher frequency of femoral neck fracture in KF patients than in the general population [84]. Similarly, Fusaro et al. observed a significant risk of fracture with elevated phosphate levels in CKD patients [85]. The direct role of phosphate in the pathogenesis of MBD was demonstrated in experimental data, which observed that inorganic phosphate induces apoptosis of cultured osteoblast-like cells and inhibits RANK–RANKL-signaling-mediated cell differentiation of cultured osteoclast-like cells [86,87].

## 4. Management of Hyperphosphatemia

Many studies have underscored the association of hyperphosphatemia with increased risk of CVD, MBD, and mortality [3,5,6,7,43,44,45]. Furthermore, despite the known association of hyperphosphatemia and poor outcomes, there has been a steady rise in mean serum phosphate concentrations amongst KF patients in the United States [43,88]. Hence, it appears logical to control serum phosphate and its associated hormonal perturbations in individuals with CKD and KF. The most recent Kidney Disease Improving Global Outcome (KDIGO) guidelines recommend lowering elevated phosphate levels toward the normal range [89]. Current management strategies include removal of phosphate by dialysis, reduction in dietary phosphate intake, and reducing intestinal absorption of phosphate.

### 4.1. Removal of Phosphate by Dialysis

As discussed in an earlier section, hyperphosphatemia and its metabolic consequences in KF are initiated by the inability of kidneys to adequately excrete phosphate. Hence, dialysis, as kidney replacement therapy, is utilized as a tool to remove phosphate from blood with a hope to mitigate hyperphosphatemia related adverse outcomes. However, both traditional in-center hemodialysis (HD) and peritoneal dialysis (PD) are grossly inadequate for phosphate removal (Table 1).

As most phosphorus is stored in bone or intracellular space, only 1% of total body phosphorus is available in extracellular space and accessible for removal by traditional HD. Consequently, in traditional three times a week dialysis, phosphate is mainly removed from extracellular fluid and with minimal removal from intracellular or tissue pools [90]. In an earlier study, DeSoi and Umans found a rapid early phosphate removal with nadir of serum phosphate around 2 h followed by a rapid rebound to 88–100% of their pre-dialysis serum phosphate within 4 h post-dialysis [91]. Similar results were observed in other studies that revealed a rapid decline in serum phosphate during the early phase of HD, followed by plateauing of serum phosphate in the final 2–4 h of the treatment [8,83,84,85,86,87,88,89,90,91,92,93,94,95,96]. In the ensuing post-HD period, phosphate fluxes from intracellular compartments and bones back to the extracellular compartment, thereby equilibrating the serum phosphate level [8,83,84,85,86,87,88,89,90,91,92,93,94,95,96]. All the aforementioned studies did not control dietary measures or use of phosphorus binders. Interestingly, in a recent study comprising 13 HD subjects, who had been off phosphate binders for 10 days and consumed a standardized low phosphate (900 mg/day) diet for 3 weeks prior to the assessments, there was an average drop in serum phosphate of −2.85 mg/dL post-dialysis. Furthermore, ‘rebound’ or return to baseline levels occurred slowly during the 24 to 48 h, suggesting that the low dietary intake prolonged the return of the serum phosphate level to baseline in the post-dialysis period [97].

While standard HD is inadequate in effective phosphorus removal, other HD modalities may be more efficient in the management of hyperphosphatemia. For instance, compared to standard HD, more frequent short daily HD or nocturnal HD are shown to be more effective in serum phosphate management, likely due to rate of phosphate moving from different physiological compartments and continued removal of phosphate from ultrafiltration [98]. Indeed, convective clearance is more effective in phosphate removal, as shown by Minutolo and colleagues in a single blind crossover study of 12 patients undergoing two different dialysis treatments. One dialytic treatment utilized both diffusive and convective fluid fluctuations using post-dilution reinfusions of bicarbonate at high rates of ultrafiltration in one of the three treatments per week. A standard bicarbonate hemodialysis treatment was performed as the other study treatment. They observed higher removal of inorganic phosphate in patients undergoing the combined diffusive and convective hemodialysis treatment compared to those undergoing standard HD alone [99].

Compared to HD, phosphate removal in PD is more complex. Unlike HD, where the dialysis membranes have well-defined and relatively uniform characteristics, the peritoneal membranes vary among different patients and in the same individual over dialysis vintage [100]. Hence, the transfer rate of solutes including phosphate vary among different individuals and depend upon peritoneal membrane permeability and dialysis prescription (number of exchanges, dwell time, dialysate composition, and ultrafiltration) [101]. Moreover, compared to other small solutes of similar molecular weight, phosphate peritoneal clearance is lower [102], underscoring the important role of peritoneal solute transfer characteristics and PD prescriptions in influencing phosphate removal and, therefore, serum phosphate control. In a prospective observational study including 380 adult peritoneal dialysis patients, Courivaud C. et al. [101] observed that a slower peritoneal transporter status was associated with reduced weekly peritoneal phosphate clearance and consequently higher serum phosphate levels.

Additionally, irrespective of peritoneal phosphate transfer rate, patients receiving continuous ambulatory peritoneal dialysis (CAPD) with long dwell time showed a greater peritoneal phosphate clearance compared to patients treated with automated peritoneal dialysis (APD) with shorter dwell time. Accordingly, peritoneal phosphate removal could be enhanced by increasing dialysate volume and dwell time in CAPD. In APD, phosphate clearance can be increased by increasing volume, number, and duration of dialysis cycles as well as by addition of longer daytime exchanges. Similarly, in an observational cross-sectional study, Debowska M. et al. [103] assessed phosphate clearance by CAPD, continuous cyclic peritoneal dialysis (CCPD), and APD. Patients treated with CAPD showed a greater total weekly phosphate removal compared to CCPD and APD as well as lower serum phosphate level compared to APD patients. In a similar fashion, in a recent multicenter prospective cohort study conducted on 737 patients, transition from CAPD to APD was followed by an increase in serum phosphate, whereas the opposite occurred after switching from APD to CAPD, underscoring the importance of long dwell times in the management of serum phosphate levels [104]. Therefore, notwithstanding overall inadequate phosphorus removal by PD, individualized PD prescriptions are important, taking into account the peritoneal phosphate transfer rates, especially in slow transporters and particularly those with low residual renal function undergoing APD [101].

### 4.2. Dietary Management of Hyperphosphatemia

A reduction in kidney function promotes hyperphosphatemia due to impaired phosphate excretion in the setting of ongoing dietary phosphate intake. Hyperphosphatemia is independently associated with dire consequences, including bone and mineral disease, high risk of CV disease, and mortality [105,106,107]. Hence, it is imperative to address the dietary phosphorus intake to minimize these complications associated with hyperphosphatemia and increase a better outcome in patients with CKD and KF [108].

The typical daily phosphate intake in a Western diet is about 1500 mg, of which about 1200 mg is absorbed in the gastrointestinal tract, depending upon the bioavailability of phosphate in food (Figure 1) [109]. Based upon the current guidelines, patients with KF are typically recommended a daily phosphorus intake of 900 mg/day [110]. Furthermore, the guidelines highly emphasize considering the source of phosphorus in the dietary recommendation for hyperphosphatemia [110]. Traditional approaches for dietary phosphorus restriction can compromise both nutritional adequacy and quality of life. The challenge lies in creating a diet that is both nutritionally adequate, palatable, cost effective, and promotes adherence with minimal impact on quality of life (QOL). Hence, understanding the different dietary forms of phosphorus and their bioavailability is essential for dietary planning in KF patients.

## 5. Forms of Phosphorus in the Diet: Organic and Inorganic

Phosphorus in the diet exists primarily in two forms: organic and inorganic. Organic phosphorus is predominantly found in plant- and animal-based food sources and is bound to proteins, lipids, or other organic molecules. Phosphorus from animal sources, which is mainly present as phosphoproteins and phospholipids, is more bioavailable than plant-derived phosphorus [111]. In plants, phosphorus is primarily stored as phytate (inositol hexaphosphate), which is poorly absorbed by humans due to the absence of the enzyme phytase, which is required for its breakdown [105,106]. In contrast, inorganic phosphorus is commonly found in food additives, preservatives, supplements, and pharmaceutical agents and exists as free phosphate ions (Table 2) [106].

The distinction between these forms is crucial because their bioavailability, or the percentage absorbed in the gastrointestinal tract, varies significantly. Inorganic phosphorus is more readily absorbed than organic phosphorus, which has important implications for phosphorus balance, particularly in individuals with CKD and KF [105]. Specifically, inorganic phosphorus has a bioavailability of 80–100%, while organic phosphorus is absorbed at rates of 30–60% from animal sources and 20–40% from plant sources (Table 2) [111].

## 6. Intervention

In clinical practice, dietary recalls and food diaries are typically utilized to estimate dietary phosphate intake in patients with CKD and KF. However, a considerable amount of hidden phosphorus is present in preservatives, additives, and pharmaceutical preparations that is not often recognized by individuals and healthcare providers (Table 2) [107,112]. These hidden sources can contribute a considerable portion of daily phosphorus intake (up to 500–1000 mg) [112]. Moreover, inorganic phosphates are nearly 100% absorbed in the gastrointestinal tract, making them a major contributor to hyperphosphatemia.

The American food industry plays a major role in exacerbating phosphorus-related issues in CKD patients due to its heavy reliance on processed foods and additives. Unlike many other countries, the U.S. diet is dominated by ultra-processed foods, fast food, and convenience meals, all of which contain high levels of inorganic phosphorus from preservatives and additives such as sodium phosphate [113]. In contrast, many other countries emphasize whole, minimally processed foods, which contain primarily organic phosphorus from plant and animal sources, with considerably lower absorption rates. Moreover, one of the challenges in tracking phosphorus intake is the lack of transparency in food labeling. The US Food and Drug Administration (FDA) does not require phosphorus content to be listed on nutrition labels, making it nearly impossible for patients and dietitians to accurately track intake [114]. Even when food labels list the phosphorus content, they fail to account for the amount and bioavailability. As a result, tracking phosphorus intake becomes even more complicated, potentially leading to gross underestimation of actual intake.

Furthermore, inorganic phosphate is added to medications and nutrition supplements and is another, generally unrecognized, source of phosphate exposure. In a Canadian hemodialysis population, 11% of prescribed medication contained a phosphate salt with a median phosphate burden of 111 mg per day [115]. With a median daily pill burden of 19, prescription medications can substantially contribute to the daily phosphate load in dialysis patients [116]. Moreover, phosphate is usually added to multivitamin supplements with estimations between 20 and 150 mg per supplement. The use of a multivitamin supplement will further contribute to a higher dietary phosphate load.

A diet that reduces phosphorus intake while maintaining nutritional balance is imperative (Table 1). The source of phosphorus, as well as its bioavailability, is important to consider when incorporating dietary changes. As discussed above, the bioavailability of phosphorus in plant-based foods is much lower than that in animal-based foods. Thus, the use of plant-based foods may help to reduce phosphorus levels more effectively while promoting nutritional adequacy [108]. In an experimental study with rats, it was shown that a plant-based diet significantly lowered phosphorus levels compared to an animal-based diet [117]. Likewise, in a study in human subjects with CKD stages 3–4, those individuals who followed a vegetarian diet exhibited lower serum phosphate levels, reduced serum FGF-23 levels, and less urinary phosphate excretion compared to those on a meat-based diet containing the same amount of phosphate [118]. Even a partial replacement of animal-based with plant-based sources significantly lowered serum phosphorus levels in patients with CKD [119]. Furthermore, data from both prospective and retrospective cohort studies indicate that people following a plant-based diet, including vegetarians, had significantly reduced serum phosphate levels in cases of kidney failure [120,121].

The critical role of plant-based diets in CKD and KF management has been underscored by recent international guidelines. The International Society of Renal Nutrition and Metabolism (ISRNM) recommends incorporation of plant-based protein sources into the dietary plans of CKD/KF patients while considering individual preferences and cultural dietary habits [122]. Likewise, the 2024 KDIGO Clinical Practice Guideline for the Evaluation and Management of CKD emphasizes a balanced, diverse diet, prioritizing plant-based foods while limiting animal proteins and processed foods [123]. Hence, a well-planned diet should minimize processed foods and additives and emphasize plant-based sources. This can not only help control phosphorus load but also addresses comorbidities such as diabetes, hypertension, and metabolic acidosis, which are prevalent among CKD and KF patients [108,124,125]. Additionally, low-protein diets can slow kidney function and reduce the risk of kidney failure in non-dialysis CKD patients [126,127].

One of the concerns of strict dietary phosphate restriction is the risk of malnutrition. In contemporary practice, patients with KF are often discouraged from consuming plant-based proteins and instead focus on animal-based proteins. This recommendation is based on the misperception that plant-based protein sources have a high phosphorus content and can lead to hyperphosphatemia. In reality, the bioavailability of phosphorus is not appropriately acknowledged. While animal-based foods are dense in protein, the phosphorus content is highly bioavailable, whereas plant-based foods have a high phosphorus content but low bioavailability. While both types of proteins contain phosphorus, a considerably smaller amount is absorbed from plant-based foods. The aforementioned studies have shown that plant-based proteins actually help improve phosphorus load [108,127,128,129]. Furthermore, a plant-based diet in patients with kidney failure is associated with improved nutritional status [130]. Moreover, a plant-based diet is richer in micronutrients such as vitamins and minerals, and it is also linked to better-controlled metabolic acidosis. In conclusion, consuming a plant-based diet is much more effective in controlling hyperphosphatemia than consuming an animal-based diet.

Dietary modifications for managing kidney disease can be both costly and challenging for patients to maintain [131]. These specialized diets often require careful planning and can be more expensive than standard dietary options, potentially leading to financial strain [131]. In contrast, processed foods are typically more affordable and convenient, aligning with the fast-paced nature of American lifestyles. However, it is important to recognize that while processed foods may offer short-term savings, they can contribute to long-term health issues, potentially increasing healthcare costs over time [130]. Plant-based diets, such as vegan or vegetarian options, can be more cost-effective in the long run. For instance, a recent modeling study comparing food prices from the international comparison program for 150 countries showed that healthy and sustainable dietary patterns were up to 25–29% lower in cost [132]. Variants of vegetarian and vegan dietary patterns were generally more affordable, and vegan diets reduced food costs by up to one-third compared to meat and dairy products [133]. Moreover, a low-fat vegan diet has been associated with a 16% decrease in total food costs, offering potential economic benefits alongside health improvements [134]. However, it is crucial to note that the affordability of plant-based diets can vary based on geographic location, availability of fresh produce, and individual dietary needs [135]. In some cases, specialized plant-based products may be more expensive, offsetting potential savings [135]. In summary, while dietary modifications for health conditions can be costly and require substantial adherence, plant-based diets may offer a balance between health benefits and cost savings [136,137]. Furthermore, dietary counseling can focus on other simple food preparation techniques that can be utilized to lower dietary phosphate content. For instance, boiling reduces phosphate content due to the demineralization of food, as the minerals move from the food into the boiling water [132]. Reported phosphate reductions by boiling vary between 35% and 50% [138].

Medical nutrition therapy (MNT) comprising dietary education and counseling by a registered dietitian (RD) is crucial in the management of hyperphosphatemia [139]. Indeed, the 2020 KDOQI Clinical Practice Guideline for Nutrition in CKD update recommends MNC as an important strategy to optimize nutritional status, and to minimize risks associated with comorbidities and metabolic derangements in patients with CKD and KF [126]. Likewise, KDIGO underscores the role of renal dietitians in formulating dietary modifications according to disease severity and comorbidities, addressing sodium, phosphorus, potassium, and protein intake [123]. Key components of effective dietary management include careful planning, regular assessment of nutritional status, and monitoring adherence. CKD patients often face comorbidities requiring specific dietary management, which can be overwhelming [140]. The dietitian’s role goes beyond advice, providing individualized, holistic counseling based on the patient’s health, preferences, and pre-existing conditions. Effective education, motivation, and alternative food options tailored to the patient’s preferences are essential for adherence [141]. It is important for patients to be active participants in the management of their dietary modifications, and the dietary advice should be based upon a shared decision making between the patient and their care team.

In conclusion, dietary phosphate restriction in CKD patients requires a rational approach rather than an indiscriminate prescription of reduced dietary protein intake. The nature of dietary phosphate, including hidden sources, should be carefully examined and avoided (Table 1). Appropriate dietary counseling and educational programs involving self-care and shared decision making are vital. Encouraging patients to reduce meat consumption and to shift to a grain-based vegetarian diet may allow for sufficient protein intake without adversely affecting serum phosphate. Furthermore, it must be stressed that dietary phosphate restriction is difficult to accept and usually insufficient to achieve adequate control of serum phosphate. Therefore, other strategies, such as pharmacological measures, and phosphate removal by dialysis in patients with KF, should be discussed when deemed necessary.

### 6.1. Reducing Intestinal Phosphate Absorption

While dietary restriction and dialysis are able to reduce phosphorus levels to a certain extent, most individuals with KF need additional pharmaceutical interventions targeting small intestinal phosphate absorption for the management of hyperphosphatemia. The current pharmacological approaches include use of phosphate binders and intestinal phosphate transport inhibitors (Table 1).

Phosphate binders: More than 80% of patients with KF take one or combination of phosphate binders [142]. These agents bind with dietary phosphates to form insoluble complexes in the intestinal lumen and excreted in the feces [143,144,145,146,147]. Phosphate binders can be calcium or non-calcium based. Calcium-based binders in common use include calcium carbonate and calcium acetate. Non-calcium-based binders include now-defunct aluminum-based binders, resin-based binders (sevelamer), and metal-based binders (lanthanum carbonate, sucroferric oxyhydroxide, and ferric citrate).

Aluminum is an avid phosphate binder and was the pillar of hyperphosphatemia management in the early years. However, its use has been largely abandoned after realization of systemic toxicity due to its accumulation in the brain, bone, and bone marrow, causing encephalopathy, dementia, osteomalacia, and anemia, respectively [148,149,150]

Calcium-based Binders: After retraction of aluminum use, calcium-based binders became the predominant agents for hyperphosphatemia management in KF [151,152]. The avidity to bind phosphate per gram of calcium and capacity to lower serum phosphate is similar between calcium carbonate and calcium acetate [152,153,154]. While there was no evidence of a difference in the risk of hypercalcemia between the two agents, concern for risks of hypercalcemia and vascular calcification with the use of calcium-based binders remains a matter of concern. Whereas studies comparing outcomes of calcium-based binders with non-calcium binders, mainly sevelamer, have yielded mixed results, the overall findings from these studies seemed to show either a potential for benefit or an absence of harm associated with calcium-free phosphate-binding agents compared with calcium-based agents [154,155,156,157,158,159,160,161]. Based upon these studies, KDIGO-2017 guidelines recommend restricting the use of calcium-based binders in the management of hyperphosphatemia in KF [89]. Aside from hypercalcemia, the major adverse events associated with calcium-based agents are gastrointestinal symptoms, which are similar between calcium carbonate and calcium acetate and less frequent than those associated with sevelamer [154,155,156].

Resin-based Binders: Sevelamer-based binders are the original non-calcium-based phosphate binders and currently the most often used phosphate binder in clinical practice [162,163,164,165,166]. Sevelamer is a cationic polymeric ion exchange resin that binds to dietary phosphate without being degraded or absorbed, thereby suppressing intestinal phosphate absorption. [167]. It was first released as sevelamer hydrochloride but given concerns about metabolic acidosis due to the hydrochloride moiety, the formulation was later changed to sevelamer carbonate, which appears to have a similar effect on phosphate lowering [168,169]. While sevelamer is effective in reducing blood phosphorus levels, high pill burden, and gastrointestinal side effects may lead to non-adherence, and consequent poor control of serum phosphate level [170]. Over 25% of individuals experience adverse effects, including constipation, abdominal discomfort, nausea, and dyspepsia [168,171]. Rarely, sevelamer can cause lower gastrointestinal bleeding due to deposition of sevelamer crystals in colonic mucosa and ensuing mucosal damage [172,173].

In addition to chelating phosphate, sevelamer binds bile salts, resulting in a significant reduction in serum total cholesterol and low-density lipoprotein cholesterol, but may concurrently interfere with the absorption of the fat-soluble vitamins A, D, E, and K and other nutrients [174,175,176].

#### 6.1.1. Metal-Based Phosphate Binders

Lanthanum carbonate, a chewable, calcium-free metal cation that has minimal intestinal absorption, was approved in 2004 for phosphate chelation. While its efficacy of phosphate binding is similar to those of other binders, one advantage of lanthanum carbonate is lower pill burden compared with those of previous phosphate binders [177,178,179]. In a multicentered randomized controlled trial in non-dialysis CKD patients, lanthanum was observed to be effective in reducing phosphate levels [180]. Likewise, in another study randomizing HD patients to calcium carbonate and lanthanum groups, phosphate levels fell similarly between the two groups, while hypercalcemia was limited to calcium carbonate groups [181]. Key adverse effects of lanthanum reported in a systematic review included vomiting, diarrhea, intradialytic hypotension, cramps, myalgia, and abdominal pain [182].

Based on the experience with aluminum-based preparations, buildup of heavy metals in the body is a cause of concern. In fact, accumulation of lanthanum in the liver and many other organs has been reported in uremic rats [183]. While lanthanum is present in the lysosomes of hepatocytes, there are no reports of cells or tissue damage in the liver and no increase in the incidence of adverse events associated with any organ function, including liver after up to 6 years of treatment in KF patients [184]. Moreover, lanthanum also accumulates in the bones of chronic dialysis patients, with a 50- to 80-fold increase in bone content after 1 to 3 years of lanthanum carbonate therapy [184,185]. However, paired bone biopsies confirmed no accumulation of lanthanum in patients treated with lanthanum carbonate over a long period of time [186].

#### 6.1.2. Iron-Based Binders: Ferric Citrate and Sucroferric Oxyhydroxide

Ferric citrate is a new type of phosphate binder that binds phosphate in exchange to citrate to form insoluble ferric phosphate, which is excreted in the feces, thereby lowering the serum phosphate level. In a phase III RCT that compared ferric citrate with non-iron-containing phosphate binders (sevelamer or calcium-based), ferric citrate was noninferior to sevelamer or calcium-based binders in controlling serum phosphate [187]. Additionally, it increased serum ferritin, reduced the need for intravenous iron and erythropoietin- stimulating agents, and improved overall anemia management [187,188]. Furthermore, ferric citrate reduced FGF23 levels and lowered the PTH level, thereby exerting effect on secondary hyperparathyroidism [189]. One concern commonly raised regarding use of ferric citrate in KF patients is risk of aluminum toxicity, as citrate can potentially enhance the absorption of aluminum. However, a RCT comparing the safety profile of ferric citrate with control group (sevelamer and calcium acetate) in KF patients on HD did not observe aluminum toxicity among recipients of ferric citrate [190].

Sucroferric oxyhydroxide is a chewable phosphate binder, comprising a mixture of polynuclear iron-oxyhydroxide, sucrose, and starch. After oral administration, the sucrose and starch are broken down, releasing polynuclear iron-oxyhydroxide that binds to phosphates in the intestinal lumen to form an insoluble compound. One advantage of the chewable sucroferric oxyhydroxide is its ability to disintegrate rapidly upon contact with water or saliva [191]. Sucroferric oxyhydroxide has high phosphate-binding capacity across the physiological gastrointestinal pH range [192]. Moreover, in the phase III RCT comparing sucroferric oxyhydroxide and sevelamer in PD and HD patients, similar efficacy of serum phosphate reduction was observed with the two agents [193,194]. However, sucroferric oxyhydroxide reduced the pill burden by 75% and consecutively improved treatment adherence [194]. Adverse effects are primarily gastrointestinal, including diarrhea, nausea, constipation, and vomiting [194]. Similar results were observed in another study that randomized CKD patients to sucroferric oxyhydroxide and sevelamer. There was a significant and sustained 30% reduction in serum phosphate and a significant 64% decrease in FGF-23 at 24 weeks with sucroferric oxyhydroxide. Furthermore, there was a significant reduction in PTH level at week 24, but it returned to nearly the baseline level at week 52 [195]. In the post hoc analysis and extension of the phase III trial [194], compared to sevelamer, there was a significant increase in transferrin saturation and the hemoglobin level with sucroferric oxyhydroxide in the first 24 weeks from the baseline; ferritin level also trended up but did not achieve statistical significance [196]. Conversely, no significant changes in iron-related parameters were noted over the 6-month of observation in another study [197].

With availability of a wide variety of phosphate binders, the choice of binder for an individual patient has become difficult. The ideal phosphate binder should be inexpensive, have low pill burden, efficient in binding dietary phosphate, and safe, with minimal systemic absorption and few side effects. Unfortunately, none of the binders discussed above meet all these criteria. In a recent network meta-analysis examining 22 different strategies and 16 different phosphate-lowering agents, sevelamer was noted to be the most common binder used clinically [162]. Moreover, all investigated drugs exhibited superior or comparable efficacy in phosphorus reduction compared to placebo [162]. The current KDIGO guidelines recommend restricting calcium-based binders and considering non-calcium binders due to concern about calcium load and consequent vascular calcification with the former [89]. As there is no convincing outcome data comparing one non-calcium binder with another, the choice of non-calcium binders can be difficult and should consider the cost, side effects, pill burden, and collateral benefits, if any. For instance, ferric citrate may assist with anemia management, sucroferric oxyhydroxide and lanthanum carbonate may provide a lower pill burden, and sevelamer may have low-density lipoprotein-lowering and anti-inflammatory effects [175,177,178,179,187,188,194].

#### 6.1.3. Drugs Inhibiting Intestinal Phosphate Transport

As discussed in earlier sections, intestinal phosphate absorption occurs via two distinct pathways: the 1,25 dihydroxy vitamin D-dependent saturable transcellular pathway through NaPi-IIb and the non-saturable paracellular pathway that is dependent upon NHE-3 [11,13,14,15,16]. Moreover, intestinal phosphate chelation may result in compensatory upregulation of NaPi-IIb [157]. Therefore, inhibition of phosphate transporter activities is a plausible alternative or complementary approach to reduce phosphate load in KF patients (Table 2) [198].

Nicotinamide (also known as niacinamide) is a form of vitamin B3 that reduces intestinal NaPi-IIb expression and can potentially lower sodium-dependent intestinal phosphate absorption [199,200]. However, recent randomized trials have demonstrated limited efficacy of nicotinamide in reducing phosphate level, poor tolerance, and some safety concerns [201,202]. For instance, in an RCT, nicotinamide as an add-on therapy to phosphate binders in HD patients did lower phosphate level at 24 weeks, but the effect was not maintained at 52 weeks [202]. Moreover, nicotinamide was associated with higher rates of side effects, including diarrhea, pruritus, and thrombocytopenia. Hence, the use of nicotinamide as a sole agent for the management of hyperphosphatemia is not recommended.

Tenapanor is an inhibitor of the NHE3 that reduces paracellular intestinal sodium and phosphate absorption [23,24,26,203]. In a phase-III randomized, double-blind placebo-controlled trial, 8-week treatment with tenapanor significantly reduced serum phosphate by a mean of 1.0–1.2 mg/dL in hyperphosphatemic patients on HD [26]. Furthermore, in another double-blind phase-III trial, 4-week treatment with tenapanor as an add-on therapy to binders achieved a larger mean change in serum phosphate compared to binders plus placebo in HD or PD patients who were hyperphosphatemic despite receiving phosphate binder therapy [167]. In both studies, adverse events were mainly restricted to stool softening and increased bowel movements resulting from increased stool sodium and water content due to inhibition of intestinal NHE-3 by tenapanor [26,203,204]. In October 2023, the US FDA approved tenapanor as an add-on therapy for use in adult dialysis patients who have an inadequate response to phosphate binders or who are intolerant of any dose of phosphate binder therapy [205].

## 7. Controversies and Challenges in the Management of Hyperphosphatemia

The current paradigm in the management of hyperphosphatemia in KF focuses on reduction of the serum phosphate level principally by dietary phosphate restriction and pharmaceutical measures to diminish intestinal phosphate absorption. Indeed, KDIGO-2017 guidelines recommend lowering phosphate to the normal range in people with KF [89]. Consequently, rather than using hard endpoints, clinical trials for phosphate-lowering agents use reduction in serum phosphate level as the primary end point [26,169,180,181,187,193,194,195,202,203,204]. Similarly, regulatory agencies approve these drugs based upon their capacity to reduce phosphate level and not on the outcome data [205]. Adherence to the guidelines-recommended phosphate target often requires colossal efforts that include substantial dietary restrictions and considerably increase pill burden at a considerable cost. Generally, individuals with KF take loads of prescription medications. Indeed, in a cross-sectional study the median daily pill burden of 19 was noted among KF patients [206]. Phosphate binders accounted for 49% of this pill burden [206]. Similarly, in a Japanese dialysis cohort phosphate binders comprised 33% of total pill burden [207]. High pill burden and the need to take the binders with every meal and snack can have considerable psychosocial consequences and impact quality of life [206,208]. While the pill burden can be reduced with the appropriate choice of binders, add-on therapy with new agents, such as tenapanor as well as appropriate dietary measures and cost of novel agents, may be prohibitive. Indeed, phosphate-lowering therapies represent a sizable financial burden, with an annual cost totaling $1500 per user and $658 million aggregate for the prevalent dialysis population [4]. Furthermore, like pharmacologic measures, dietary phosphate restriction can be cumbersome. KF patients, in addition to watching phosphorus, need to adhere to myriads of dietary restrictions, such as sodium, potassium, calories (as half of KF individuals have diabetes), and fluid intake. Moreover, strict phosphate control may limit necessary protein intake. Such arduous restrictions run the risk of malnutrition, which is associated with lower survival in KF. This can be mitigated with dietary counseling, avoiding non-protein sources of phosphorus, using plant-based proteins that contain phosphate of low bioavailability, and diligent use of phosphate-lowering medications.

Given immense efforts and financial resources devoted to lower serum phosphate level, it would be logical to presume that there are high-powered studies demonstrating improvement in health outcomes in KF persons with a tight control of hyperphosphatemia. However, such data are unfortunately lacking. The 2017-KDIGO recommendations to lower phosphate levels are based upon low-quality-grade 2-C evidence [89]. Moreover, the appropriate target phosphate level for optimizing bone health and improving cardiovascular risk without causing unintended consequences remains vague. Two recent large RCTs have tried to define the specific phosphate targets. The recently halted HiLo trial defines tight control as <5.5 mg/dL, while the ongoing PHOSPHATE trial (NCT03573089) lowers this cutoff closer to 4.5 mg/dL and should provide some guidance in the ensuing years [209,210].

While managing serum phosphate levels, it should be realized that development of overt hyperphosphatemia is a late event in CKD, preceded by a chain of adaptive events, including elevated FGF-23, PTH, and low 1,25 dihydroxy-vitamin D levels, which have been independently associated with high CV risks, BMD, and mortality. Conceivably, deleterious effects associated with hyperphosphatemia could be attributable to disruptions in phosphorus-sensing mechanisms and hormonal imbalance thereof, rather than with overt hyperphosphatemia. By the time hyperphosphatemia ensues, CV disease may already be far advanced, and reduction of hyperphosphatemia is inadequate in mitigating CV risks and mortality, akin to the ineffectiveness of traditional CV risk factors in reducing CV mortality in CKD [211]. Further exploration of mechanisms that precisely control phosphorus sensing and regulation may facilitate the development of strategies to diminish the deleterious effects of phosphorus load early in the CKD course, before overt hyperphosphatemia is evident to improve overall outcomes in KF. Sans high quality evidence, tight phosphate control at the expense of QOL may not be justified. Therefore, until then, management of hyperphosphatemia should be based upon shared decision making between the individuals, dieticians, physicians, and other care providers. It should focus not only on clinical outcomes such as comorbidities and other clinical issues related to CKD/KF but also consider patient-oriented goals, such as patient preference, QOL, nutritional status, pill burden, and financial constraints.

## Figures and Tables

**Figure 1 nutrients-17-01587-f001:**
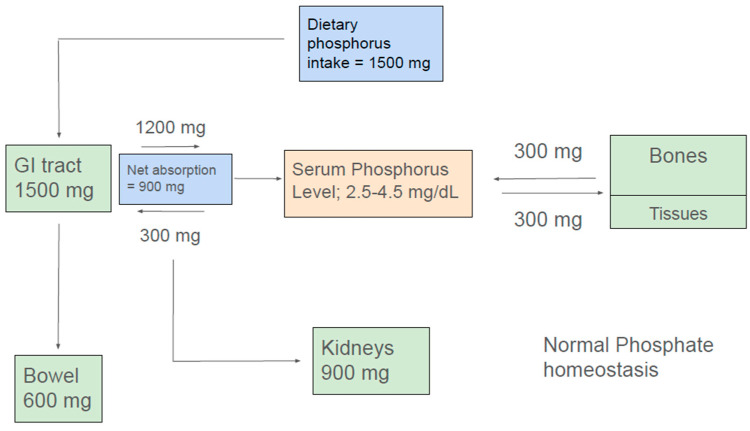
Normal phosphate homeostasis.

**Figure 2 nutrients-17-01587-f002:**
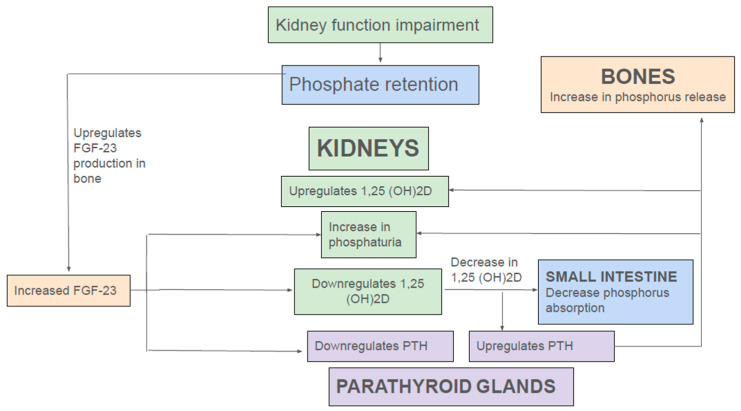
Endocrine regulation of phosphate in kidney disease.

**Table 1 nutrients-17-01587-t001:** Current strategies to manage hyperphosphatemia in chronic kidney disease and kidney failure.

**Dietary Interventions**	**Comments**
Limit processed meats, processed cheese, and dairy products	Contain highly bioavailable (80–100%) inorganic phosphates
Choose fresh meat or fish without added phosphates	Contain organic phosphates of intermediate bioavailability (40–60%)
Choose plant based food (Like legumes, soy products, nuts, whole grains)	Contains organic phosphate complexed with phytates. Has low bioavailability (20–40%)
Be aware of hidden phosphates in prescription/ over-the counter medicines and supplements	May contain considerable amount of highly bioavailable (80–100%) inorganic phosphates
**Reduction of Intestinal Phosphate Absorption**	
**Phosphate Binders**	**Comments**
Calcium acetate: Daily dose 1334–2001 mg)	High pill burden. Risk of calcium load and vascular calcification. Cheap
Calcium carbonate: Daily dose 1250–3750 mg)	High pill burden. Risk of calcium load and vascular calcification. Cheap
Sevelamer hydrochloride and sevelamer carbonate: Daily dose 2400–9600 mg	High pill burden. Gastrointestinal side effects. More expensive than Ca-based binders
Lanthanum carbonate: Daily dose 1500 mg	Low pill burden. Chewable tablets or powder preparations. Gastrointestinal side effects. No evidence of hepatotoxicity
Sucroferric oxyhydroxide: Daily dose 7.5–15 g	Low pill burden. Chewable. Low iron absorption. Primary gastrointestinal side effects.
Ferric citrate: Daily dose 630–1260 mg	Low pill burden. High iron absorption. May help with anemia management. Primary gastrointestinal side effects. Risk of aluminum toxicity as citrate increases absorption of aluminum
**Intestinal Phosphate Transport Inhibitors**	**Comments**
Nicotinamide	Blocks small intestinal active transport of phosphate via NaPi-IIb. Limited efficacy and poor tolerance due to side effects (Diarrhea, pruritus, and thrombocytopenia). Not recommended for hyperphosphatemia management
Tenapanor: Dose: 10–30 mg twice a day	Inhibits small intestinal paracellular transport of phosphate by blocking sodium/hydrogen exchanger 3. Used as an add-on therapy with phosphate binders. The major adverse effect is an increase in stool frequency and diarrhea
**Removal of Phosphate by Dialysis**	
**Dialysis Modality**	**Comments**
In-center hemodialysis 3-times a week	Inadequate in removing daily phosphate load. Patients need additional measures (diet and phosphate lowering agents) to manage hyperphosphatemia
Short-daily or nocturnal hemodialysis	Much better in controlling hyperphosphatemia
Peritoneal dialysis	Inadequate in removing daily phosphate load. Patients need additional measures (diet and phosphate-lowering agents) to manage hyperphosphatemia

**Table 2 nutrients-17-01587-t002:** Dietary sources of phosphorus.

Sources of Phosphate	Nature of Phosphate	Bioavailability	Examples
Plant-based foods	Organic Phosphates Complexed With Phytates	20–40%	Nuts and seeds Legumes Whole grain Leafy greens Soy products
Animal-based foods	Organic Phosphates	40–60%	Chicken/Poultry Red meat Fish/ sea food Milk/dairy products
Preservatives/additives	Inorganic Phosphates	80–100%	Soft drinks Processed foods Canned foods
Medicines/supplements	Inorganic Phosphates	80–100%	OTC * multivitamins/supplements Prescription medicines

* OTC: Over-the-counter.

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
