# Peer review of "Hyperphosphatemia in Kidney Failure: Pathophysiology, Challenges, and Critical Role of Phosphorus Management"

_nutrients, 2025, doi:10.3390/nu17091587_

Round 1
Reviewer 1 Report
Comments and Suggestions for Authors
The article sent for my review is very interesting, and it is clear that the authors are experts. Although it is not a systematic review, I think the authors should write what databases they used, what keywords they used, what period of publication of the studies they took into account. Whether they excluded any articles.
In addition, I wondered whether it would not be worth adding here the aspects of phosphates and klotho, see the effect of oxidative stress on calcium and phosphate disorders in CKD figure 4 in https://doi.org/10.3390/antiox9080752.
Besides, you can find relevant articles in
https://doi.org/10.1096/fj.08-123992, doi.org/10.3389/fendo.2020.00560, https://doi.org/10.1016/j.redox.2021.102173
10.1053/j.ajkd.2024.02.008
The authors mentioned this klotho protein in one sentence, and it seems to me that this is a bit too little, considering that klotho is a protein that is still being discovered.
A list of abbreviations at the end of the article would be useful.
I suggest adding something like a graphical abstract.
In general, I consider the article to be exceptionally interesting, carefully prepared with great potential, which will certainly interest Nutrients readers.
Author Response
Thank you for your kind comments. We appreciate your review of our manuscript.
Comment 1: Although it is not a systematic review, I think the authors should write what databases they used, what keywords they used, what period of publication of the studies they took into account. Whether they excluded any articles.
Response 1: Thank you for your comment. We agree that while it is not a systemic review, we have used the following databases to search for the articles used in this review and we have added the details of our search in the manuscript. There were no exclusions and we searched the databases until March 20, 2025.
Comment 2:
In addition, I wondered whether it would not be worth adding here the aspects of phosphates and klotho, see the effect of oxidative stress on calcium and phosphate disorders in CKD figure 4 in https://doi.org/10.3390/antiox9080752.
Besides, you can find relevant articles in
https://doi.org/10.1096/fj.08-123992, doi.org/10.3389/fendo.2020.00560, https://doi.org/10.1016/j.redox.2021.102173
10.1053/j.ajkd.2024.02.008
The authors mentioned this klotho protein in one sentence, and it seems to me that this is a bit too little, considering that klotho is a protein that is still being discovered.
Response 2: Thanks for your encouraging comment. We agree that klotho plays a key role in the pathogenesis of hyperphosphatemia and bone and mineral disease. We have now added two sections on pages 4-5 (Endocrine Regulation of Phosphate Homeostasis in Health and Kidney Disease, Hyperphosphatemia: Cardiovascular risks and mortality) on klotho, describing its role in relation to the pathogenesis of hyperphosphatemia and metabolic consequences. We have also utilized the references that were kindly provided by the reviewer.
Comment 3: A list of abbreviations at the end of the article would be useful.
Response 3: We have added a list of abbreviations to the manuscript.
Comment 4: I suggest adding something like a graphical abstract.
Response 4: We have added a graphical abstract in a powerpoint format.
Reviewer 2 Report
Comments and Suggestions for Authors
I considered the manuscript entitled “Hyperphosphatemia in Kidney Failure: Pathophysiology, Challenges and Critical Role of Phosphorus Management” by
Swetha Raju and Ramesh Saxena, that is intended to be published in Nutrients Journal.
The review describes everything you wanted to know about phosphorus in kidney failure and were afraid to ask. I mean it is a complete and comprehensive manuscript concerning the issue where almost every knowledge about the matter is comprised, using a language that is easy to understand, especially for clinicians. Particularly interesting is the description of nowadays status of the new drug, tenapanor which is an inhibitor of the NHE3 that reduces paracellular intestinal sodium and phosphate absorption. The rest of medications and dietary restrictions are old acquaintances, but authors introduce several studies that ponder the importance of phosphorus as central player among mineral bone disease, cardiovascular risk and kidney function.
In some paragraphs the text appears repetitive of previous paragraphs, some further editing is recommended
The abstract is not an abstract ad hoc. It is just too long summary of all the text. Readers should only read this abstract and does not need to read the full text. Clearly, it should be shortened.
Author Response
Thank you for your kind comments. We appreciate your review of our manuscript.
Comment 1: In some paragraphs the text appears repetitive of previous paragraphs, some further editing is recommended
Response 1: We have now edited those sections to avoid repetition.
Comment 2: The abstract is not an abstract ad hoc. It is just too long summary of all the text. Readers should only read this abstract and does not need to read the full text. Clearly, it should be shortened.
Response 2: We have shortened the abstract.
Round 2
Reviewer 2 Report
Comments and Suggestions for Authors
no further